# Bioactives Overproduction through Operational Strategies in the Ichthyotoxic Microalga *Heterosigma akashiwo* Culture

**DOI:** 10.3390/toxins15050349

**Published:** 2023-05-20

**Authors:** Adrián Macías-de la Rosa, Miguel Ángel González-Cardoso, María del Carmen Cerón-García, Lorenzo López-Rosales, Juan José Gallardo-Rodríguez, Sergio Seoane, Asterio Sánchez-Mirón, Francisco García-Camacho

**Affiliations:** 1Department of Chemical Engineering, University of Almeria, 04120 Almeria, Spain; amd202@ual.es (A.M.-d.l.R.); mgc459@ual.es (M.Á.G.-C.); llr288@ual.es (L.L.-R.); jgr285@ual.es (J.J.G.-R.); asmiron@ual.es (A.S.-M.); fgarcia@ual.es (F.G.-C.); 2Research Centre Ciambital, University of Almeria, 04120 Almeria, Spain; 3Department of Plant Biology and Ecology, University of the Basque Country (UPV/EHU), 48080 Bilbao, Spain; sergio.seoane@ehu.eus; 4Technology and Research Centre for Experimental Marine Biology and Biotechnology (PiE-UPV/EHU), 48013 Bilbao, Spain

**Keywords:** microalgae, *Heterosigma akashiwo*, PUFAs, fucoxanthin, bioactives, culture mode, downstream, photobioreactor

## Abstract

The red tide-forming microalga *Heterosigma akashiwo* has been associated with massive events of fish deaths, both wild and cultured. Culture conditions are responsible for the synthesis or accumulation of some metabolites with different interesting bioactivities. *H. akashiwo* LC269919 strain was grown in a 10 L bubble column photobioreactor artificially illuminated with multi-coloured LED lights. Growth and production of exopolysaccharides, polyunsaturated fatty acids (PUFAs), and carotenoids were evaluated under different culture modes (batch, fed-batch, semicontinuous, and continuous) at two irradiance levels (300 and 700 µE·s^−1^·m^−2^). Continuous mode at the dilution rate of 0.2·day^−1^ and 700 µE·s^−1^·m^−2^ provided the highest production of biomass, PUFAs (132.6 and 2.3 mg·L^−1^·day^−1^), and maximum fucoxanthin productivity (0.16 mg·L^−1^·day^−1^). The fed-batch mode accumulated exopolysaccharides in a concentration (1.02 g·L^−1^) 10-fold over the batch mode. An extraction process based on a sequential gradient partition with water and four water-immiscible organic solvents allowed the isolation of bioactive fucoxanthin from methanolic extracts of *H. akashiwo*. Metabolites present in *H. akashiwo,* fucoxanthin and polar lipids (i.e., eicosapentaenoic acid (EPA)), or probably such as phytosterol (β-Sitosterol) from other microalgae, were responsible for the antitumor activity obtained.

## 1. Introduction

The Raphidophyceae are flagellated unicellular microalgae that live in diverse brackish, marine, and freshwater habitats. Three freshwater genera are recognised (*Gonyostomum*, *Merotricha*, and *Vacuolaria*), and seven ones from seawater (*Chattonella*, *Chlorinimonas*, *Fibrocapsa*, *Haramonas*, *Heterosigma*, *Psammamonas*, and *Viridilobus*) [1]. Marine raphidophytes are typically recognised as ichthyotoxic organisms since several species have been associated with finfish kills, such as *Chattonella marina*, *C. antiqua*, *C. ovata*, *C. subsalsa*, *Fibrocaspa japonica*, and *Heterosigma akashiwo*. Toxicity mechanisms by these raphidophyceans are not fully understood. Production of brevetoxin or brevetoxin-like compounds was reported for *C. antiqua*, *C. marina*, *F. japonica*, and *H. akashiwo* [2,3]. Some species of raphidophytes are known to produce toxins in controlled cultures [4], and this ability to produce toxins can be used to obtain target commercial biomolecules [5,6].

Toxicity associated with raphidophytes has been attributed to the production of two reactive oxygen species (ROS), superoxide (O^2−^) [7,8] and hydrogen peroxide (H_2_O_2_) [9]; haemolytic compounds, which lyse red blood cells [10,11,12]; and a brevetoxin-like organic neurotoxin [3]. However, none of these toxic agents have been independently linked to the ichthyotoxicity of *H. akashiwo*. Studies reveal that not all blooms of *H. akashiwo* are toxic [13]. When *H. akashiwo* is grown under nutrient-limited conditions, it generally exhibits increased toxicity due to changes in metabolomic dynamics caused by cell stress [14].

*H. akashiwo* has been used to produce high-quality biodiesel, achieving a total lipid content higher than several other microalgal species traditionally used for this same purpose [15,16,17]. This microalga can tolerate wide fluctuations in temperature, salinity, and nutrient conditions [18], being able to proliferate in wastewater commonly derived from agriculture [19] or even under certain levels of herbicides such as glyphosate [20]. *H. akashiwo* is also an excellent candidate for growing using industrial emissions (flue gases), since this alga can metabolise nitric oxide (NO) into cellular nitrogen via a novel chimeric protein, NR2-2/2HbN [21]. Moreover, the production of polyunsaturated fatty acids, such as EPA or DHA, normally accounts for 46–50% of the total fatty acids for this microalga [22]. The fatty acid 18:5ω3 was detected in *H. akashiwo*, which has traditionally been associated with species among Dinophyceae and Prymnesiophyceae classes; it being in *H. akashiwo* is the first report of this acid in a Raphidophycea [22]. *H. akashiwo* is also a source of fucoxanthin in a higher proportion, but also zeaxanthin, violaxanthin, or β-carotene, among others [23]. On the other hand, optimal growth and establishing conditions to enhance bioactive compound production are among the difficulties in culturing raphidophyte microalgae. *H. akashiwo* has been successfully cultured in different culture systems, such as flasks from 0.1 L to 4 L [7,19,21,24], but most of these studies were assayed in small volumes indoor conditions. Only two studies of scale-up have been reported for this microalga, both carried out in bubble column photobioreactors (PBRs); one of them was to obtain biomass as feedstock for producing biodiesel [16], whereas the other one was to produce anaesthetic bioactives [25]. These wide conditions and different purpose studies suggest that this microalga would be a viable option for commercial-scale culture.

In this work, *H. akashiwo* was cultured in bench-scale bubble column photobioreactors under different culture modes and irradiance levels. Productions of biomass, exopolysaccharides, and other high-added-value compounds (carotenoids and PUFAs) were assessed. An extraction process based on a sequential gradient partition with water and four water-immiscible organic solvents with increasing polarity (hexane, toluene, dichloromethane, and n-butanol) was applied to separate target compounds.

## 2. Results and Discussion

### 2.1. Operational Conditions in the LED-Based Bubble Column PBR

Figure 1A depicts the evolution of the biomass concentration (Cb) over time for different culture modes: batch (B), fed-batch (FB1, FB2, FB3), semicontinuous (S), and continuous (C). The batch cultivation mode started at a concentration of 0.02 g·L^−1^. After a three-day lag phase, a sustained growth stage began, reaching a concentration of 0.2 g·L^−1^ after 20 days. Subsequently, the first fed-batch (FB1) culture reached a biomass concentration of around 0.6 g·L^−1^ on day 26. However, from that day, the Cb began to decline. Then, stock of f/2 medium was added again on day 30, beginning the fed-batch (FB2). During FB2, the growth rate slowed down compared to FB1. Therefore, a new fed-batch (FB3) was started right after adding tripled f/2 medium stock (f/2 × 3) and increasing irradiance I_omax_ up to 700 µE·s^−1^·m^−2^. No photoinhibition was observed. It is in line with previous studies reporting robust growth of *H. akashiwo* at 1200 µE·s^−1^·m^−2^ [26]. From day 37 to 51, FB3 conditions favoured rapid biomass growth, the culture reaching a maximum concentration of around 1 g·L^−1^. However, from day 52, growth declined. Thus, on day 58, the semicontinuous mode (S) was set by removing a variable culture volume and replenishing with an equal volume of fresh f/2 × 3 medium, the starting biomass concentration being the same as that of FB1. In this way, it could be observed if a higher concentration of nutrients and higher irradiance resulted in a higher growth rate. However, the observed concentration evolution was similar or even lower. In this way, the LC269919 strain was not limited by light or nutrients in FB1. A higher concentration of nutrients would only be necessary to sustain a higher concentration in the culture. Finally, a continuous mode growth mode was initiated by setting a dilution rate of 0.2 days^−1^. For almost two weeks, a steady-state biomass concentration was attained with an average value of 0.6 g·L^−1^ (see Figure 1A).

Figure 1B shows the average and maximum biomass productivities reached in each culture mode assayed. It can be observed that there was a considerable difference between continuous mode and batch and fed-batch modes. For continuous mode, maximum and average productivities were 132.6 and 128.2 mg·L^−1^·day^−1^, respectively. On the other hand, the batch mode registered the lowest productivity, with values of 12.6 and 6.8 mg·L^−1^·day^−1^, respectively. Productivity values reached in fed-batch modes almost tripled that of batch mode; in semicontinuous mode, the productivity was 4-fold over batch mode; productivities attained in the continuous mode were 18-fold those of batch mode. If fed-batch with high and low concentrations of nutrients are compared, there was hardly any difference in maximum values, although in the averages, the difference was 30%.

Despite numerous publications reported on cultures with *H. akashiwo*, only a few studies were carried out at the pilot scale [15,25]. Typical productivity values for *H. akashiwo* described in the literature for non-agitated or bubbled batch cultures are 47–54 mg·L^−1^·day^−1^ (strain CCMP 2393) [26] and 200 mg·L^−1^·day^−1^ (Strain ICMB 252) [15], respectively. It is difficult to make comparisons because they were obtained for different strains and culture media, modes, and even cultivation systems. However, it seems clear that intensive systems achieve higher productivity. As for the cultivation modes, the continuous system operating at a high concentration is the most productive system in terms of biomass but resulted in being not stable over time.

### 2.2. Influence of Operational Conditions on PUFA Profile and Content

The majority of saponifiable fatty acids found in this Radophycea are 14:0, 16:0, 16:1n7,18:4n3 (SDA), 20:5n3 (EPA), and 22:6n3 (DHA) [7]. In this work, we compare the three culture modes, finding the highest amount of fatty acids accumulated in biomass was found in fed-batch culture mode (FB2), exceeding by almost 20% the content compared to batch mode and almost 30% compared to continuous mode (Figure 2A). Regarding the distribution of saturated fatty acids (SFAs), monounsaturated fatty acids (MUFAs), and PUFAs (Figure 2B), they remained unchanged in the three groups, being around 38–50% for SFAs, around 15–20% for MUFAs, and 27–42% for PUFAs. As shown earlier [7], similar results for this species have been obtained, mainly in the MUFA proportion. However, in our experimental conditions, we obtained SFA content higher than PUFAs due to the higher irradiance (maximum 700 µE·m^−2^·s^−1^) compared to the value at which these authors carried out their cultures (150 µE·m^−2^·s^−1^). It is well known that in situations of high irradiances, SFAs accumulate to the detriment of PUFAs [27]. The highest productivity of the three grouped fatty acids was obtained for the continuous culture since it was the culture mode that presented the highest biomass productivity. It is known that as biomass productivity increases, the productivity of fatty acids does as well [28].

### 2.3. Influence of Operational Conditions on Carotenoid and Exopolysaccharides Content

Fucoxanthin and violaxanthin are considered by diverse authors as the major carotenoids found in *H. akashiwo* and as marker pigments for this species. As shown earlier [29], fucoxanthin, violaxanthin, antheraxanthin, zeaxanthin, and β-carotene in the biomass of *H. akashiwo* have been detected. By contrast, other authors [18] did not find zeaxanthin and antheranthin, likely because their cultures were not photoinhibited, the condition under which the photochemical epoxidation of the violaxanthin cycle occurs in cells. In our results, we find that in the profile of carotenoids exists in three carotenoids (violaxanthin, anteraxanthin, and zeaxanthin) at the same time. In the fed-batch mode, the total carotenoid content accumulated was more than 0.6% d.w 2.3-fold over the continuous mode and 1.4-fold over the batch. The fucoxanthin content was the major carotenoid, reaching its maximum value of 0.35% d.w. (Figure 3A). The maximum productivity of fucoxanthin was accumulated in continuous mode (0.17 mg·L^−1^·day^−1^) (Figure 3B), because the biomass productivity is maximum in this cultivation mode (Figure 1B). The fucoxanthin and PUFAs share antioxidant activity. Fucoxanthin is an antenna pigment, and it is accumulated in no high irradiance levels, while PUFAs are structural lipids and primary metabolites associated with the growth, and their level also decreases in high content of irradiance. Both can be oxidised in high-level irradiance [27]. The fed-batch mode accumulated exopolysaccharides in a concentration (1.02 g·L^−1^) 10-fold over the batch mode. Our data reinforce the hypothesis that carbohydrates are the preferred energy storage compounds in *H. akashiwo* when neither light nor CO_2_ is limiting [26]. Carbohydrate accumulation is a more efficient energy storage strategy because less ATP and NADPH per carbon are required for carbohydrate synthesis versus TAG synthesis [30].

### 2.4. Influence of Operational Conditions on Growth and Sequential Extraction Liquid–Liquid on Bioactivity Assays as Antiproliferative against Human Tumour Cells

Figure 4A,B show the results of the recovery of both fatty acids and carotenoids in the different stages of extraction. Figure 4A shows a recovery of almost 90% of fatty acids distributed in three stages. In the first stage, 80% of the total content was recovered. On the other hand, a homogeneous distribution of fatty acids between hexane and toluene solvents is observed, indicating that fatty acids are distributed between neutral lipids and polars in this strain. However, it is observed that saturated fatty acids were in a higher proportion in the most apolar fraction (hexane) as opposed to the polyunsaturated acids (SDA and EPA) that were found in the most polar fraction (toluene) (Figure 4C). As for the total recovery of carotenoids (Figure 4B), 82% of the total content was recovered. This was distributed in the first extraction (78%) and in subsequent extractions (4%). For both fatty acids and carotenoids, it may not be economical to use a two-stage process to recover an additional 4–5%. In the case of carotenoids, the content of polar carotenoids (xanthophylls) was the majority of the apolar (carotenoids) content. It is noteworthy that the totality of xanthophylls (fucoxanthin as the majority), except for the proportion of zeaxanthin, was extracted by toluene. β-carotene was located entirely in hexane, being more apolar. Therefore, we propose a method of separation of fucoxanthin in toluene and β-carotene in hexane. Brevenal and brevetoxins were not detected in the extracts from *H. akashiwo*. Similar observations were shown in [25]. However, antiproliferative bioactivity against different human tumour cells (See M&M) was measured in all extracts except from that obtained with hexane (Figure 5A–E). The toluene fraction was active due to the presence of fucoxanthin and polar lipids, specifically SDA or EPA; similar results were found [31]. On the other hand, the fractions of dichloromethane and n-butanol were the most active, in which neither fatty acids nor carotenoids are detected, so the greater bioactivity may be attributed to more polar compounds such as phytosterols. This is consistent with results reported on similar metabolites present in *H. akashiwo* (fucoxanthin and polar lipids such as phytosterol from other microalgae were responsible for antitumor activity obtained [32,33]) [34,35]. Bioactivity can also be observed caused by exopolysaccharides or anaesthetic bioactives that would be extracted with n-butanol [25]. However, a more comprehensive study of elucidation and separation of molecules is necessary to assign the bioactivity to the latter by RMN and mass spectrometry in future work.

## 3. Conclusions

Cultures of *Heterosigma akashiwo* were carried out in 10 L bubble column PBRs under different modes of operation. The use of continuous cultivation mode shows a high biomass productivity maintained for almost 20 days, as well as accumulation of PUFAs. The application of fed-batch mode with concentrated nutrients (f/2 × 3) showed an increase in the percentage of fucoxanthin present in biomass. Likewise, a mechanism of extraction of compounds by liquid–liquid sequential extraction using solvents of different polarity managed to effectively separate the pigments fucoxanthin and β-carotene from the rest of the carotenoids located in the fraction of toluene and hexane, respectively. In view of the results, most of the compounds of interest (carotenoids and fatty acids) were collected in the first extraction (close to 80%), successive extractions not being necessary. The extracts obtained from this fractioning were tested against various human tumour cell lines reaching high percentages of inhibition for the phases extracted from dichloromethane, n-butanol, and to a lesser extent, toluene.

## 4. Materials and Methods

### 4.1. The Microalgae, Inoculum, and Its Maintenance

*H. akashiwo* strain BMCC75 [36] microalgae was provided by the Basque Microalgae Culture Collection (BMCC) from the University of the Basque Country (UPV/EHU) (LC269919–GenBank accession number) (Figure 6A). Inocula were maintained in 2 L flasks under 12/12 h light–dark illumination regimen at 18 ± 2 °C; the illumination was provided by 32 W fluorescent lamps rendering an average irradiance of 100 µE·m^−2^·s^−1^ on the culture flask surface. The culture medium was prepared from natural, filter sterilised (0.22 μm Whatman GF/F 47 mm, Maidstone) Mediterranean seawater with 30 psu. The preparation of the culture medium was made with the f/2 formulation [37] at a molar N:P ratio of 5.

### 4.2. The LED-Based Bubble Column PBR

*H. akashiwo* was photoautotrophically cultured in bubble column PBRs (Figure 6B). The PBR was made up of a clear column constructed from poly(methyl methacrylate) with an internal diameter of 86 mm, a height of 2 m, and a wall thickness of 2 mm. Throughout all conditions, the culture height (H), excluding the volume fraction of the dispersed air phase (i.e., gas hold-up), was set at 1.73 m, equivalent to a working culture volume of 10 L. To mix the culture broth, sterile filtered air was bubbled into the vessel through a 2 mm diameter nozzle located at its base. An air flow rate of 0.04 *v/v*·min was set to prevent the cells from suffering hydrodynamic stress. Illumination was provided by multicolour LED strips (red, green, blue, and warm white, collectively referred to as RGBWW; Edison Opto Co., Taiwan) attached in a helical way vertically with a nut pitch of 7 cm, as seen in Figure 6B, using a total of 7.5 m of LED strip per bubble column. A 12:12 h light–dark cycle was imposed, in which maximum irradiance at midday (I_omax_) was assayed to two levels, 300 and 700 µE m^−2^ s^−1^. Each PBR contained a U-shaped stainless steel heat exchanger with an internal diameter of 6 mm and a total length of 3.5 m, through which thermostatic water was circulated using a minichiller inverter H6 (MUENR-14-H6T) in order to maintain a controlled culture temperature of 21 ± 1 °C. The pH was controlled at pH 8.5 by automatically injecting carbon dioxide as needed. The fresh culture medium was prepared using filter-sterilised Mediterranean seawater. Before starting the series of experiments, each PBR was sterilised with sodium hypochlorite solution and subsequently neutralised with sodium thiosulfate, as described earlier [38]. A total of 8 L of the medium was inoculated with 2 L of an inoculum containing microalgal cells in the late exponential growth phase inoculated with an initial concentration culture of 0.02 g·L^−1^.

At the beginning of the experiment (stage B), the culture was grown in batch mode with the f/2 formulation and at a value of I_omax_ of 300 µE m^−2^ s^−1^. In the second stage (FB1), the culture was implemented with f/2 stock equivalent to a nutrient concentration in the medium of f/2. In the third stage (FB2), fed-batch culture continued to re-supplement nutrients as in FB1. In the fourth stage (FB3), conditions of the fed-bath culture were changed by increasing I_omax_ at 700 µE m^−2^ s^−1^ and the addition of f/2 stock equivalent to a nutrient concentration in the medium of f/2 × 3. The different culture mode changes were made once growth was observed to slow or cease. Next, a semicontinuous culture mode (S) at the same illumination regimen and medium formulation was set, consisting in removing a culture volume of 8 L and replenishing it with an equal volume of fresh f/2 × 3 medium. Lastly, the third culture mode was a continuous one (S), which was set up at 700 µE·m^−2^·s^−1^ and a concentration of nutrient of f/2 × 3 in feed stream into the PBRs, as pointed out in Figure 2. The dilution rate was set to 0.2 day^−1^, consisting in removing 2 L of culture daily and replacing it with the same volume of freshly prepared medium. The culture samples were collected, centrifuged at discontinuous mode (BECKMAN COULTER Allegra 25 r) at 7000× *g* 6 min with which 0.2–0.4 L of concentrated microalgae were obtained, washed with a 0.5 M ammonium bicarbonate solution [39], and then freeze-dried. The dry biomass and supernatants were immediately analysed or stored frozen at −22 °C.

### 4.3. Kinetic Parameters

Biomass productivity (Pb) expressed in mg·L^−1^·day^−1^ was calculated by dividing the change in biomass concentration (mg·L^−1^) by the time (day) elapsed during the evaluated cultivation period. The maximum biomass productivity (P_bmax_) was determined by selecting the highest value achieved during the evaluated period. The final productivities of the different compounds of interest studied (P_CARO;_ carotenoids (fucoxanthin, zeaxanthin, or β-carotene); P_PUFAs_; PUFA; P_MUFA_; MUFA; and P_SFA,_ SFA) were calculated by multiplying the biomass productivity by the percentage in dry weight of each compound.

### 4.4. Analytical Procedures

The biomass dry weight was determined with cultures between 20 and 200 mL of *H. akashiwo* cells in suspensions that were gently centrifuged (7000× *g* 6 min) in sterile 50 to 250 mL Falcon tubes. The pellets were washed with 4 to 40 mL of a 0.5 M ammonium bicarbonate solution [38], taking care not to lyse the cells, and then frozen and freeze-dried (Cryodos 50, Telstar) for 48 h. Using samples withdrawn throughout the culture and daily measurements of absorbances carried out between 300 and 800 nm, and using the value of 720 nm (OD_720_) to calculate the biomass concentration, Cb (g·L^−1^) = 1.27·OD_720_ (r^2^ = 0.97; n = 25), since this was correlated with the biomass concentration without interference from chlorophylls by dry weight.

Likewise, the ratio between the maximum variable fluorescence (Fv) and the maximum fluorescence (Fm) of the chlorophylls (Fv/Fm) was routinely measured as described in other work [40]. This value provides information about photosynthetic efficiency, a key indicator of cellular stress. These values were determined by a pulsed light (PAM) chlorophyll fluorimeter (PAM-2500 Chlorophyll fluorometer, Heinz Walz GmbH, Effeltrich, Germany).

The methods for quantifying the concentration of nitrates (4500 N) and phosphates (4500 P) present in the supernatant have been described elsewhere [41]. Nutrient measurements were made by duplicates in the supernatants.

The carotenoid content and profile were determined using an HPLC photodiode array detector, as previously explained [42], with 5 mg of lyophilised biomass. The quantified carotenoid pigments comprised fucoxanthin, violaxanthin, auraxanthin, zeaxanthin, and β-carotene. Standards of β-carotene and fucoxanthin from Sigma Chemical Co. (St. Louis, MO, USA) and violaxanthin and zeaxanthin from DHI (Hørsholm, Denmark) were used. With each of these standards, external calibration lines were established to be able to quantify the concentrations.

Fatty acid (FA) contents and profiles were obtained with 10 mg of lyophilised biomass by direct transesterification and gas chromatography (6890 N Series Gas Chromatograph, Agilent Technologies, Santa Clara, CA, USA), as described in [43].

The presence of brevetoxins (PbTx and BTX) and brevenal in the extracts was analysed according to Reference [25] using a Waters XEVO TQ-XS tandem quadrupole atmospheric pressure (API) mass spectrometer (Waters, Eschborn, Germany) equipped with a high-performance ZSpray dual-orthogonal multi-mode (ESI/SPCI/ESCi) source, coupled to a Waters Acquity UPLC system consisting of a Waters Acquity UPLC I-Class Solvent Manager, a Waters Acquity UPLC I-Class flow-through-needle (FTN) Sample Manager, and an Acquity UPLC I-Class column manager (CM-A).

EPS extraction was performed according to a previous method [44,45], modified by another author [46]. In brief, the crude EPS collection contained in supernatant (500 mL) was precipitated with 2.2 volumes of absolute ethanol at −20 °C overnight. The precipitated EPS was collected at (6000× *g* 20 min) at 4 °C. The pellet containing the EPS was dried at room temperature in a laminar flow hood for 6 hr. The EPS was introduced in a C18 sep-pack cartridge (500 mg) (Supelco) to eliminate possible salts precipitated. The dry weight of the extracted EPS was determined by drying at 105 °C to constant weight [47].

Sequential-gradient partitioning liquid–liquid extraction experiments were performed with crude methanolic extracts from 15 g of *H. akashiwo* biomass from fed-batch cultivation mode as described earlier [31].

Antiproliferative bioactivity tests were performed to evaluate the samples obtained from *H. akashiwo* against a panel of four different human tumour cell lines (i.e., ATCC^®^NSCLC A549 lung carcinoma, HT-29 ATCC^®^HTB-38 colon adenocarcinoma, MDA-MB-231 ATCC^®^HTB-26 breast adenocarcinoma, and PSN-1 ATCC^®^ CRL-3211 pancreatic adenocarcinoma). All the cells were obtained from the American Type Culture Collection (ATCC, Manassas, VA, USA), and the bioassays were performed according to [48].

## Figures and Tables

**Figure 1 toxins-15-00349-f001:**
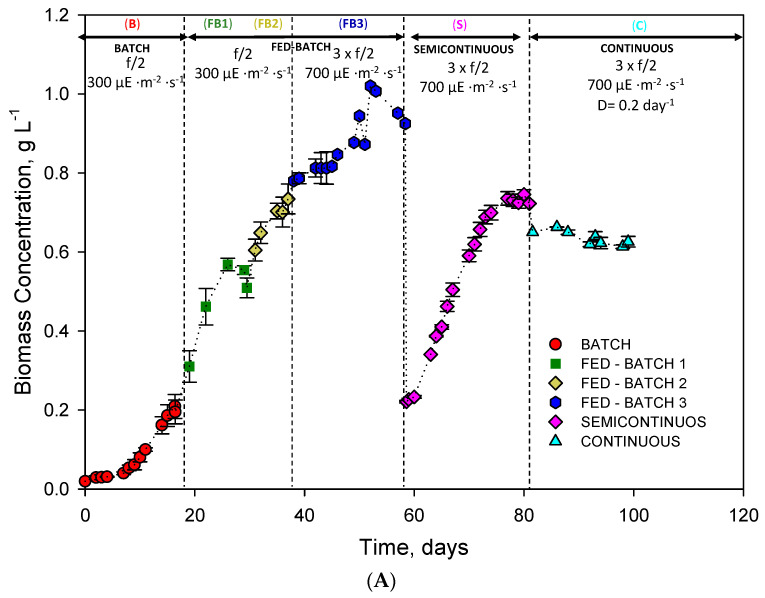
(**A**) Kinetic parameters for the microalgae *H. akashiwo* in a bubble column photobioreactor illuminated (300 to 700 µE·m^−2^·s^−1^) by LEDs expressed in g·L^−1^, using f/2 medium from ×1 and ×3. The vertical dotted lines delimit the different experimental sets performed (batch, fed-batch, semicontinuous, and continuous) for the biomass growth curves. Data points are averages, and vertical bars are standard deviations (SD) for duplicate samples. (**B**) Average biomass productivity and maximum biomass productivity calculated for all cultivation mode assayed.

**Figure 2 toxins-15-00349-f002:**
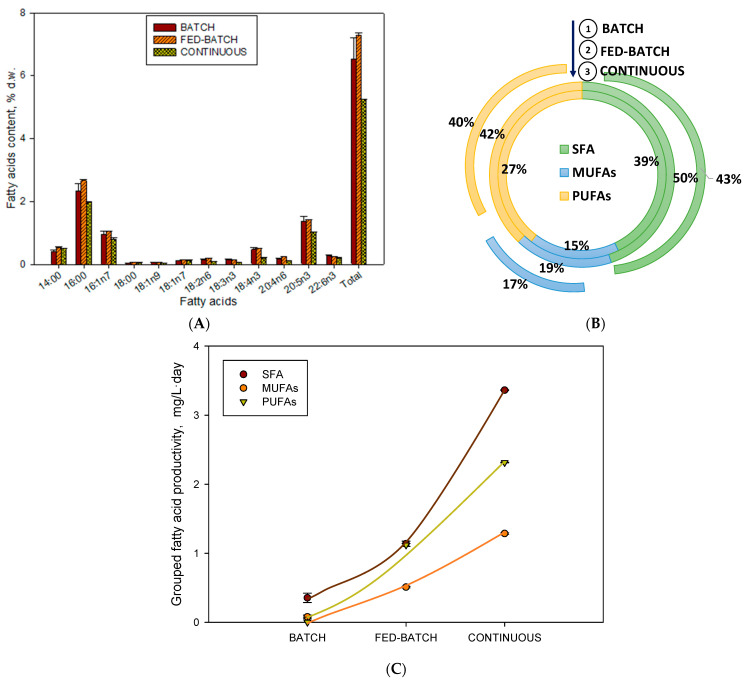
Profile and variation of the fatty acid content of the biomasses analysed for the different cultivation modes (batch, fed-batch, and continuous). (**A**) Total contents of fatty acids extracted (d.w.) with the different cultivation modes for *H. akashiwo*. (**B**) Their individual fatty acid distribution in the three fractions (Saturated (SFA), Monounsaturated (MUFAs), and Polyunsaturated (PUFAs). (**C**) Grouped fatty acid productivity for each cultivation mode (**B**, FB2, and **C**)_._ Data points and bars are averages, and vertical bars are standard deviations for triplicate samples.

**Figure 3 toxins-15-00349-f003:**
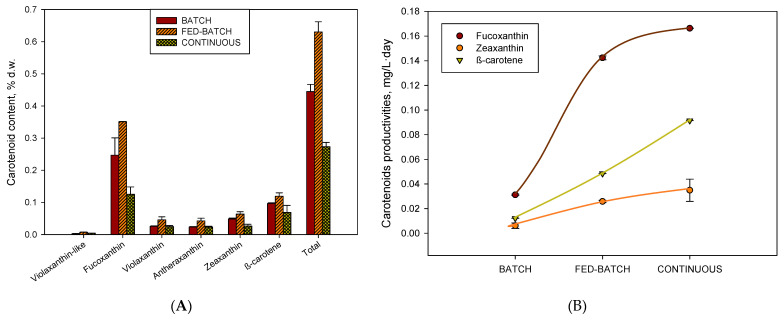
Profile and variation of the carotenoid and exopolysaccharides content of the biomasses analysed for the different cultivation modes (batch in red, fed-batch in green, and continuous in yellow). (**A**) Total carotenoids extracted (d.w.) with the different cultivation modes. (**B**) Carotenoid productivities (mg·L^−1^·day^−1^) for the different cultivation modes. (**C**) Exopolysaccharides concentration (g·L^−1^) found in the supernatant for the different cultivation modes. Data points and bars are averages, and vertical bars are standard deviations for triplicate samples.

**Figure 4 toxins-15-00349-f004:**
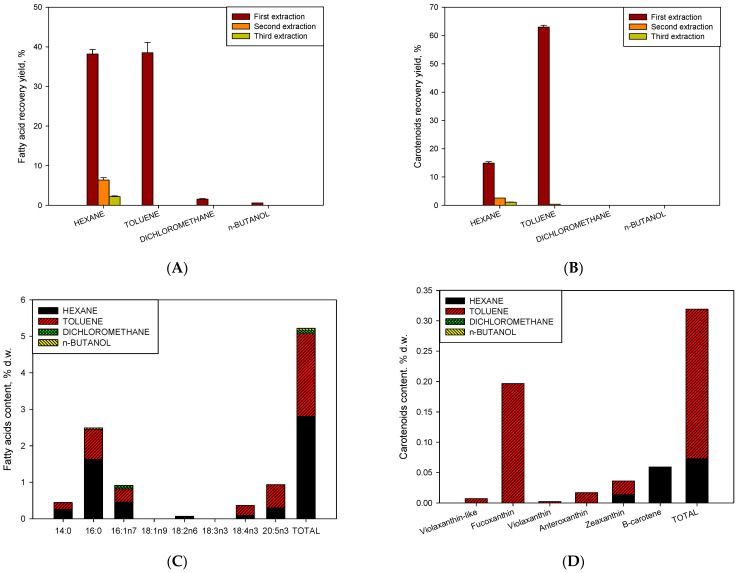
Recovery of high-value compounds obtained by sequential partitioning method for isolating bioactive compounds using hexane, toluene dichloromethane, and n-butanol in three extractions from *H. akashiwo* using a polarity gradient for fatty acids (**A**) and carotenoids (**B**). Additionally, individual fatty acids (**C**) and carotenoids (**D**) content were distributed in each solvent. Results are shown as mean ± SE (n = 3) for the first extraction in each solvent. Points without SD bars indicate that the SD was smaller than the symbol.

**Figure 5 toxins-15-00349-f005:**
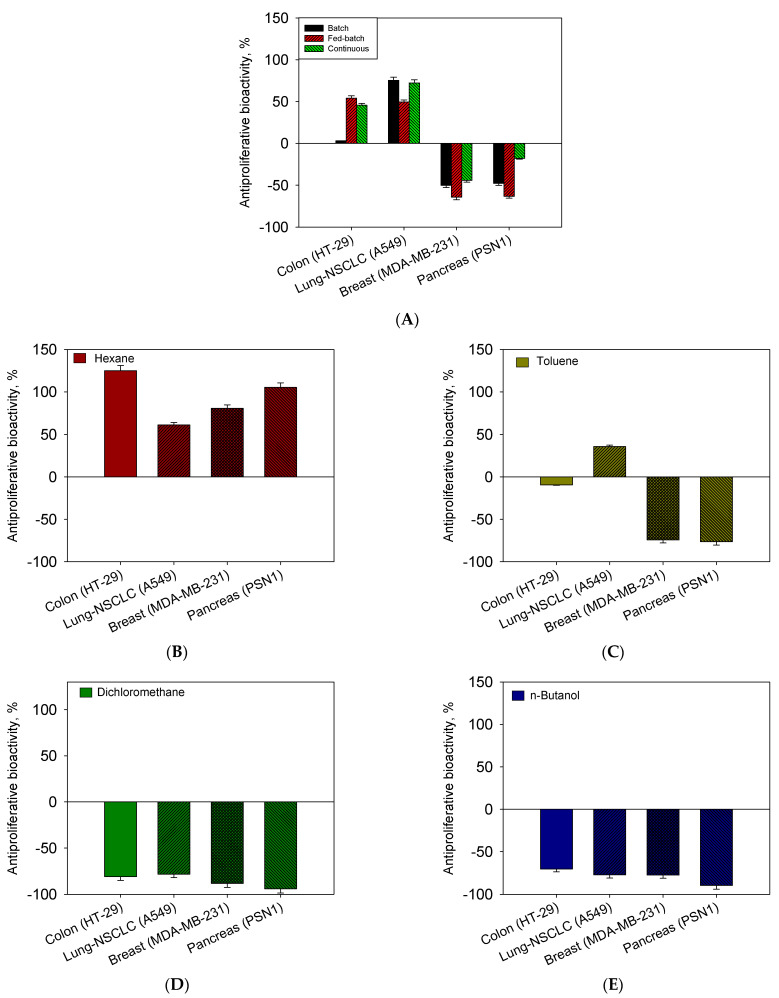
Results from the antiproliferative assays. Survival percentages of the different human tumour cell lines in extracts prepared from biomass of different culture modes and prepared from sequential partitioning by solvent polarity gradient: (**A**) biomass; (**B**) hexane; (**C**) toluene; (**D**) dichloromethane; and (**E**) n-butanol. Data points are averages, and vertical bars are standard deviations (SD) for duplicate samples. Points without SD bars indicate that the SD was smaller than the symbol.

**Figure 6 toxins-15-00349-f006:**
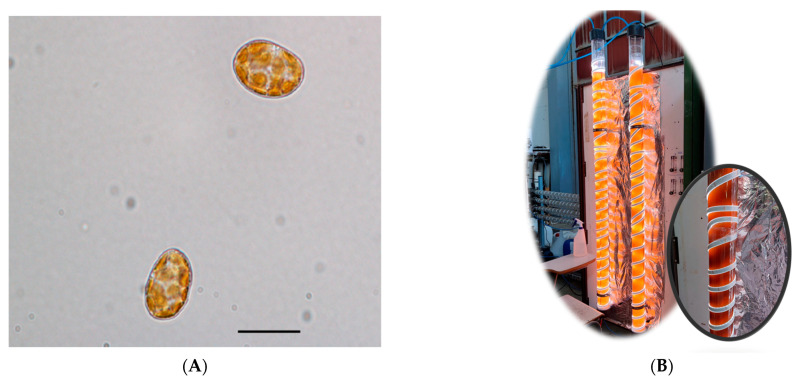
Illustration of the photobioreactor and microalgae used for the tests. (**A**). Light microscope images of living cells of *H. akashiwo* obtained at 100X (scale bar = 10 µm). (**B**) Cultures of *H. akashiwo* in the bench-scale bubble column photobioreactors used (illumination system based on strips of multicolour light-emission diodes).

## Data Availability

Not applicable.

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
