# Peer review of "Bioactives Overproduction through Operational Strategies in the Ichthyotoxic Microalga *Heterosigma akashiwo* Culture"

_toxins, 2023, doi:10.3390/toxins15050349_

Round 1
Author Response
It has been attached as a file

Reviewer 2 Report
The article "Bioactives overproduction ..." is devoted to the choice of cultivation conditions for ichthyotoxic microalga in order to obtain the maximum concentration of a number of biologically active substances.
This is important and useful work. The article describes the methods in detail. The conditions for maximum production of biomass and several biologically active substances are determined. The results obtained are presented in the form of 6 figures. Importantly, various extracts were tested against various human tumor cell lines. It was shown which extracts better inhibit the growth of tumor cells. A statistical analysis of the obtained data was carried out.
The text of ms needs more attention.
Minor edits:
1. All words in the title of an article in MDPI journals are usually capitalized.
2. H. akashiwo - with a small letter - akashiwo.
3. In the introduction, it is desirable to indicate their authors according to Algabase at the first mention of the species.
4. O2- fix to O2-.
5. Remove spaces before %.
6. “Figure” write everywhere in full.
7. “/” everywhere replace with “-1”, like L-1.
8. In the captions for the Figures, first give the name of the entire Figure, and then the values (A) and (B).
9. Avoid crediting authors when citing, it is better to write "as shown earlier [1]".
10. H. akashiwo should be written in italics throughout.
11. ... in 10-L bubble column, not 10L ...
12. Put a space before oC
13. References should be issued according to the rules of the journal.
Author Response
It has been attaches as a file
